# Changes in Exercise Capacity and Ventricular Function in Arrhythmogenic Right Ventricular Cardiomyopathy: The Impact of Sports Restriction during Follow-Up

**DOI:** 10.3390/jcm11051150

**Published:** 2022-02-22

**Authors:** Sarah Costa, Kristina Koch, Alessio Gasperetti, Deniz Akdis, Corinna Brunckhorst, Guan Fu, Felix C. Tanner, Frank Ruschitzka, Firat Duru, Ardan M. Saguner

**Affiliations:** 1Department of Cardiology, University Heart Center Zurich, CH-8091 Zurich, Switzerland; sarah.costa1193@gmail.com (S.C.); kristinakoch@bluewin.ch (K.K.); alessio.gasperetti93@gmail.com (A.G.); deniz.akdis@usz.ch (D.A.); corinna.brunckhorst@usz.ch (C.B.); guan.fu@usz.ch (G.F.); felix.tanner@usz.ch (F.C.T.); frank.ruschitzka@usz.ch (F.R.); firat.duru@usz.ch (F.D.); 2Center for Integrative Human Physiology (ZIHP), University of Zurich, CH-8057 Zurich, Switzerland

**Keywords:** ARVC, exercise, ergometry

## Abstract

(1) Background: Physical exercise has been suggested to promote disease progression in patients with arrhythmogenic right ventricular cardiomyopathy (ARVC). We aimed to investigate the exercise performance and ventricular function of ARVC patients during follow-up, while taking into account their adherence to exercise restriction recommendations. (2) Methods: This retrospective study included 49 patients (33 male, 67%) who had an exercise test at baseline and after 4.2 ± 1.6 years. Of the 49 ARVC patients, 27 (55%) were athletes, while 22 (45%) were non-athletes. Of the athletes, 12 (44%) continued intensive sports activity (non-adherent), while 15 (56%) stopped intensive physical activity upon recommendation (adherent). The maximum workload in Watts (W), percentage of the target workload (W%), and double product (DP) factor were measured for all patients. (3) Results: The non-adherent cohort had a significant decrease in physical performance (W at baseline vs. follow-up, *p* = 0.012; W% at baseline vs. follow-up, *p* = 0.025; DP-factor at baseline vs. follow-up, *p* = 0.012) over time. Left ventricular (LV) function (LV ejection fraction at baseline vs. follow-up, *p* = 0.082) showed a decreasing trend in the non-adherent cohort, while the performance of the adherent cohort remained at a similar level. (4) Conclusions: If intensive sports activities are not discontinued, exercise capacity and left ventricular function of athletes with ARVC deteriorates during follow-up. All patients with ARVC need to strictly adhere to the recommendation to cease intense sports activity in order to halt disease progression.

## 1. Introduction

Arrhythmogenic right ventricular cardiomyopathy (ARVC) is a genetically determined disease of the heart muscle, which predominantly affects the right ventricle (RV). The RV myocardial tissue is progressively replaced by fibrofatty tissue [1], which leads to RV dilatation, dysfunction, and regional wall motion abnormalities. The disease may present with life-threatening ventricular arrhythmias and sudden cardiac death (SCD) in the young, athletic population [1,2,3]. The underlying pathophysiologic mechanism in ARVC is impairment of cell adhesion [4]. Increased mechanical wall stresses that occur during endurance exercise stretch the desmosomal connections between cardiomyocytes and result in an earlier and more severe phenotypic expression [5,6].

The negative impact of exercise on ARVC was shown both in murine and clinical studies [7]. Kirchhof et al. reported that endurance exercise accelerated the development of RV dysfunction and arrhythmias in a plakoglobin (*JUP*)-deficient mouse model [8]. Likewise, there is clinical evidence that the amount and intensity of exercise increase the likelihood of diagnosing ARVC, and promote disease progression and ventricular arrhythmias not only in patients with desmosomal variants, but also in gene-elusive patients [9]. Although regular exercise is often recommended to assure cardiovascular fitness, [10,11] ARVC patients are typically advised to avoid intense physical activity, and clinical detraining has been proven beneficial in the reduction of premature ventricular contraction (PVC) burden in ARVC [12]. 

Data on the exercise capacity and ventricular function of ARVC patients over time as well as on patient adherence with regard to the expert recommendation to cease endurance and/or competitive sports is scarce. Therefore, in this study, we aimed to investigate the exercise performance and ventricular function of ARVC patients during follow-up, while taking into account their adherence to exercise restriction recommendations.

## 2. Materials and Methods

This study included patients from the multicenter Swiss ARVC Registry who fulfilled the 2010 Modified Task Force Criteria (TFC) [13] for definite or borderline ARVC and underwent multiple exercise testing. Data on baseline parameters and clinical outcomes were gathered from hospital charts. The study was approved by the Zurich Cantonal Ethical Committee (KEK-ZH-NR:2014-0443).

Analysis of exercise testing: Data of all exercise tests comprised the maximum workload in Watts (W) and as percentage of the target workload (W%), the maximum heart rate (HR), maximum systolic and diastolic blood pressure (BP), and double product (DP) factor. Physical exercise capacity was related to reference values based on age, weight, and gender. 

Patients were divided into two cohorts, according to their exercise history following the diagnosis of ARVC: an athletic cohort and a non-athletic cohort. For the athletic cohort, we considered only competitive athletes. The definition of a competitive athlete was taken from the 36th Bethesda Conference, and defined the “athlete as one who participates in an organized team or individual sport that requires regular competition against others as a central component, places a high premium on excellence and achievement and requires some form of systematic (and usually intense training)” [14]. These 2 cohorts were then divided by adherence to the recommendation to stop intensive physical activity. Data regarding exercise history was collected either per history (17 patients, 35%) or with a pre-specified patient questionnaire (32 patients, 65%) (Appendix A). The evolution of exercise capacity over time was evaluated with regard to continuation of intensive sports activity after ARVC diagnosis, and adjusted for adherence to physical exercise restrictions: in accordance with current guidelines [15] we recommended patients to stop any type of intensive physical activity and only engage in low to moderate intensity exercise. Patient adherence was assessed based on adherence to this recommendation. 

Transthoracic echocardiography (TTE): Digitally stored transthoracic echocardiographic data at the time of first and follow-up exercise testing were retrieved for each patient. Left ventricular (LV) systolic function was calculated using the Simpson’s biplane method. LV ejection fraction (LVEF) > 50% was considered normal. RV dimensions of the outflow tract (RVOT) were measured according to the 2010 TFC, while RV dilation was assessed by measuring indexed end-diastolic RV area (RVEDAi). RV systolic function was evaluated using fractional area change (FAC%) in the apical four-chamber view. 

Arrhythmic outcome: MACE (major adverse cardiac events) were defined as cardiac death, heart transplantation, sudden cardiac death, ventricular fibrillation, sustained ventricular tachycardia, including appropriate implantable cardioverter defibrillator (ICD) therapies, and arrhythmic syncope. The cut-off for a significant amount of PVCs was set at 500 PVCs/24 h, as suggested by the 2010 TFC [13].

Statistical analysis: Continuous variables were reported as [mean ± SD], if normally distributed, or as median [interquartile range] (IQR), if not normally distributed. Categorical variables were reported as counts (percentage). Comparisons between categorical variables were made using Student’s *t*-test, paired-samples *t*-test, or the non-parametric equivalents as appropriate. Comparisons between categorical variables were performed through contingency tables assessed using a *χ*^2^ or Fisher’s exact test, as appropriate. Statistical significance was set for *p* < 0.05. All p values were considered two-sided. All statistical analyses were performed using the IBM SPSS statistical package v25.0 or Stata (Ver.14.0, StataCorp, College Station, TX, USA). 

## 3. Results

### 3.1. Study Population

Forty-nine patients (33 male, 67%) were eligible for the study. The mean age at the time of first exercise testing was 46 ± 15 years. Thirty-nine (80%) of these patients had definite ARVC. Genetic testing had been performed in 39 (80%) of these patients, of whom 33 (67%) harbored a genetic variant. 24 (49%) of these variants were desmosomal, while 9 (18%) were non-desmosomal. Of the patients harboring a variant, 12 (31%) harbored a pathogenic/likely pathogenic genetic variant associated with ARVC, of which 11 (92%) were desmosomal. Mean follow-up between the first and last exercise testing was 4.2 ± 1.6 years. Thirty-four patients (69%) had an ICD, and 26 patients (53%) had at least one MACE during follow-up (Table 1). Antiarrhythmic drug use was similar between the two exercise tests (Table 2). The findings of exercise testing and transthoracic echocardiographic at baseline and during follow-up are shown in Appendix A.

### 3.2. Influence of Continued Sports Activity on Physical Performance

Of the 49 ARVC patients, 27 (55%) were athletes, while 22 (45%) were non-athletes. Of the athletes, 12 (44%) continued intensive sports activity, while 15 (56%) stopped intensive physical activity upon recommendation. In total, from the whole cohort (*n* = 49), 12 patients (24%) continued regular, intensive sports exercise (non-adherent cohort), while 37 (76%) did not engage in intensive physical activity (adherent cohort). As shown in Figure 1 and Table 2, non-adherent patients had a significant decrease in physical performance over time (W at baseline vs. follow-up, *p* = 0.012; W% at baseline vs. follow-up, *p* = 0.025; DP-factor at baseline vs. follow-up, *p* = 0.012), while the performance of the adherent cohort remained at a similar level (W at baseline vs. follow-up, *p* = 0.394; W% at baseline vs. follow-up, *p* = 0.794; DP-factor at baseline vs. follow-up, *p* = 0.852). An inter-cohort comparison (non-adherent vs. adherent) showed that, while the non-adherent cohort showed a significantly higher physical performance at baseline (W non-adherent vs. adherent, *p* = 0.024; W% non-adherent vs. adherent, *p* < 0.001; DP non-adherent vs. adherent, *p* = 0.019), exercise performance in the non-adherent cohort decreased, approaching values of the adherent cohort at the time of last follow-up, (W non-adherent vs. adherent, *p* = 0.373; W% non-adherent vs. adherent, *p* = 0.073; DP non-adherent vs. adherent *p* = 0.874).

### 3.3. Influence of Continued Sports Activity on Ventricular Function

Forty-six (94%) patients in the study cohort had a transthoracic echocardiography at baseline and forty-two (86%) had one at follow-up (Appendix A). The non-adherent cohort had a decrease in LVEF, although not significant (LVEF at baseline vs. follow-up, *p* = 0.082), whereas RV function remained stable (FAC at baseline vs. follow-up, *p* = 0.192; RVEDAi at baseline vs. follow-up, *p* = 0.969), while LVEF and RV parameters remained both stable in the adherent cohort (LVEF at baseline vs. follow-up, *p* = 0.191; FAC at baseline vs. follow-up, *p* = 0.622; RVEDAi at baseline vs. follow-up, *p* = 0.220). Furthermore, inter-cohort comparison (non-adherent vs. adherent) showed that, while at baseline the non-adherent cohort showed a significantly higher LV function (LVEF non-adherent vs. adherent, *p* = 0.019), but similar RV dimensions/function (FAC non-adherent vs. adherent, *p* = 0.778; RVEDAi non-adherent vs. adherent, *p* = 0.372), LVEF was similar between the non-adherent and adherent cohort at last follow-up (LVEF non-adherent vs. adherent, *p* = 0.579). No significant differences were observed for RV dimensions and function between non-adherent and adherent patients at last follow-up (FAC non-adherent vs. adherent, *p* = 0.100; RVEDAi non-adherent vs. adherent, *p* = 0.757) (Figure 2, Appendix A).

### 3.4. Influence of Continued Sports Activity on PVC Burden and MACE

Of the 49 ARVC patients, 42 (85.7%) had a 24 h Holter evaluation at baseline (11 non-adherent and 31 adherent), while 27 (64.3%) had a 24 h Holter evaluation at follow-up (7 non-adherent and 20 adherent). There was no difference between the two cohorts regarding the number of patients with a relevant PVC burden of >500/24 h at baseline (non-adherent 73% vs. adherent 61%, *p* = 0.496) or at follow-up (non-adherent 43% vs. adherent 60% *p* = 0.432) (Appendix A). Similarly, no significant difference was observed between the two cohorts regarding MACE (non-adherent 50% vs. adherent 54%, *p* = 0.807).

## 4. Discussion

To our knowledge, this is the first study that investigates the evolution of exercise performance and disease progression in patients with ARVC, stratified by adherence to the clinical recommendation of ceasing intensive sports activity. Our study has the following main findings: (1) At baseline, athletes with ARVC showed, as expected, a higher physical performance, as compared to non-athletic patients with ARVC. However, during follow-up exercise performance in non-adherent patients decreased, approaching values of the adherent cohort at last follow-up; (2) There was a decline in LVEF in non-adherent patients during follow-up, whereas these parameters remained stable in the adherent cohort; (3) A significant proportion (24%) of the total ARVC cohort was not adherent to the clinical recommendation to cease intensive sports activity. Of the population of athletes, 44% were non-adherent to current guideline recommendations to cease intense physical exercise, which is a remarkable finding.

Previous data on exercise performance in ARVC: Sports activity, especially at a competitive level, is an environmental factor promoting disease progression in ARVC [3], and markedly increases the risk of sustained ventricular arrhythmias and appropriate ICD shocks [7,16]. This has been demonstrated both in murine [8] and in clinical studies [7]. Moreover, it has been shown that the amount and intensity of exercise increases the likelihood of ventricular arrhythmias and heart failure in these patients [9]. Early studies assessing the impact of physical exercise on triggering ventricular ectopy have yielded controversial results [17,18]. However, exercise testing in asymptomatic carriers of ARVC-associated genetic variants was shown to expose a latent electrical substrate in these patients, by demonstrating new PVCs with a superior-axis and first-time appearance of epsilon waves [19]. Karlsson et al. studied findings of exercise testing during follow-up in ARVC patients, focusing solely on the presence of ventricular arrhythmic activity [20]. Our study differs markedly from the above-mentioned study since we have included more patients and distinguished between athletes and non-athletic ARVC patients in our cohort, and whether these patients followed the recommendations on physical exercise restrictions.

Cellular mechanisms of arrhythmia in ARVC through sports: While the exact pathogenic mechanisms by which desmosomal variants cause cardiomyocyte loss are yet to be completely understood, in recent years several groups have successfully produced in vitro cellular models of ARVC using patients human-induced pluripotent stem cell-derived cardiomyocytes (hiPSC-CM) and animal models of ARVC [8]. These studies have shown that pathogenic variants in the PKP2, JUP and DSG2 gene can cause sodium channel dysfunction and provoke ARVC. Fabritz et al. have shown in their JUP-deficient mouse model that manifestation of the phenotype including arrhythmogenicity is accelerated by endurance training [21,22]. Furthermore, cells of patients with a PKP2 variant have revealed reduced densities of the associated desmosomal protein plakoglobin and of gap-junction protein Connexin43, which are associated with a distorted structure of the desmosome, as well as prolonged field potential rise time [23,24]. Furthermore, a recent study on hiPSC-CMs derived from a patient with a DSG2variant has demonstrated that the sensitivity of the diseased cardiomyocytes to adrenergic stimulation, as occurs during physical activity, is changed in ARVC patients. This study found that Iso-induced inhibition of the sodium-calcium exchanger channel (***I*_NCX_**) and action potential duration (APD) shortening, as well as epinephrine-induced arrhythmic events were enhanced. Interestingly they also discovered multiple ion channel dysfunctions, specifically reduced ***I*_NA_**, ***I*_TO_**, ***I*_SK_**, ***I*_ATP_** and ***I*_NCX_**. The latter provides a good explanation for the Iso-induced APD-shortening, and thus provides a substrate for the occurrence of arrhythmias during exercise [25]. Thus, in ARVC patients the enhanced stress caused by increased catecholamines, as is the case during physical activity may increase the susceptibility to ventricular arrhythmias.

Influence of continued intensive sports activity on physical performance: It is known that endurance sports increases the likelihood of diagnosing ARVC and promotes disease progression [9,16]. Yet, data on exercise capacity and physical performance of ARVC patients over time is scarce. This is of clinical importance since ARVC patients are recommended to avoid intense physical activity. Indeed, it has been shown that clinical detraining is associated with decreased PVC burden. Nonetheless, athletic patients with ARVC not rarely question the recommendation to cease physical activity and want to maintain their exercise capacity by continuation of physical exercise [26]. With this regard, we have noted that a high proportion (44%) of our ARVC cohort of athletes was non-adherent to the clinical recommendation to stop intense exercise, which has not positively influenced their clinical course and exercise capacity during follow-up. We showed that exercise capacity decreased in those patients who continued intensive sports activity (non-adherent patients) as compared to those who were adherent with the recommendation to cease intensive sports activity. While the population of athletes (non-adherent), as expected, started from significantly higher exercise capacity levels, this cohort was subject to a decline of their physical fitness (decrease in maximum workload and DP factor), in contrast to the non-athletic (adherent) cohort of patients who remained stable during follow-up. Our findings are important for clinical practice. We have shown that continuation of intensive sports activity results in a more pronounced deterioration of exercise capacity, even within a mean follow-up of approximately 4 years.

Influence of continued physical activity on ventricular function, arrhythmogenicity and MACE: In our study, similar to exercise capacity, athletes (non-adherent) experienced a significant decline in LVEF, whereas LVEF remained stable in the non-athletic cohort during follow-up. This finding indicates that continuous exercise may increase the likelihood of developing heart failure and biventricular disease [27], as previously suggested by our group and others. Follow-up cardiac magnetic resonance tomography (MRI) was only available in 5 patients (*N* = 3 non-adherent and *N* = 2 adherent). Despite this low patient number, we noticed a slight increase in the RV-EDV in the *N* = 3 non-adherent patients, whereas no relevant change was observed in the *N* = 2 adherent. This is also in line with the study by Perrin et al. [19], which correlated the amount and intensity of lifetime exercise with the increase in ventricular arrhythmias and heart failure in desmosomal mutation carriers. However, changes in PVC burden (assessed by Holter ECGs) were less evident in our study. This finding may be partly due to fluctuations in PVC count over time and other external factors apart from exercise, which due to the small sample size, we could not evaluate through multivariable analysis [28]. Similarly, the lack of a difference in MACE between the two cohorts, likely stems from the small patient number, as well as the relatively short follow-up time, as this result is conflicting with previous studies [26].

Adherence: a key factor: It is important to note that a significant proportion of the total ARVC population (24%), and in particular athletes with ARVC (44%) was non-adherent to the clinical recommendation to cease intense sports exercise in this study. In a previous study, we could also show that non-adherence is an important risk factor for adverse outcomes in ARVC patients [26]. When assessing the reasons for this non-adherence, despite clear evidence for a negative impact on the disease, the most common reason was the subjective feeling of being healthy as well as fear of detraining. This interesting aspect certainly deserves further investigation in the future.

Limitations: This was a small retrospective study with associated limitations regarding patient selection, generalizability of findings, and incomplete follow-up data, especially considering the data on 24 h Holter ECG monitoring, biomarkers and cardiac MRI. Furthermore, a more precise assessment of athletic activity based on lifetime MET hours was only available in 65% of our patients. Therefore, we were not able to determine a critical cut-off for lifetime MET hours and adverse outcome in ARVC. Finally, only part of the cohort (80%) had a definite diagnosis of ARVC and genotyping was incomplete impairing robust genotype-phenotype correlation studies. Therefore, larger prospective studies are necessary to validate our findings and address these remaining questions.

## 5. Conclusions

Athletes with ARVC have a significantly higher physical performance and ventricular function as compared to non-athletic ARVC patients at baseline. However, if intensive sports activities are not discontinued, exercise capacity and left ventricular function of athletes with ARVC deteriorates during follow-up. All patients with ARVC, whether athletes or non-athletes at baseline, need to strictly adhere to the recommendation to cease intense sports activity in order to halt disease progression.

## Figures and Tables

**Figure 1 jcm-11-01150-f001:**
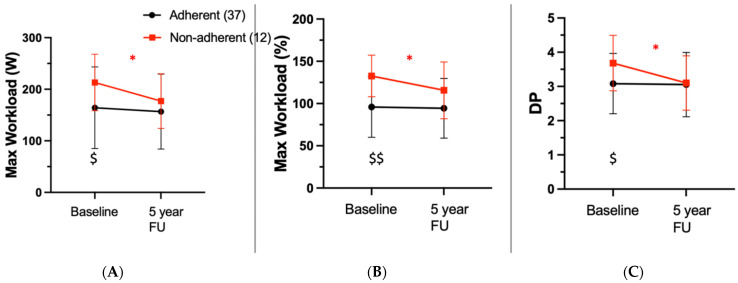
Exercise testing findings. (**A**) Comparison of maximum Workload (W) at baseline and follow-up. (**B**) Comparison of maximum Workload (%) at baseline and follow-up. (**C**) Comparison of double product (DP) factor at baseline and follow-up. Red line indicates cohort, who continued physical activity (non-adherent); black line indicates cohort, who stopped physical activity (adherent). * indicates *p* < 0.05 for paired *t*-test. $ indicates *p* < 0.05, $$ indicates *p* < 0.005 for independent Student’s *t*-tests between the two cohorts.

**Figure 2 jcm-11-01150-f002:**
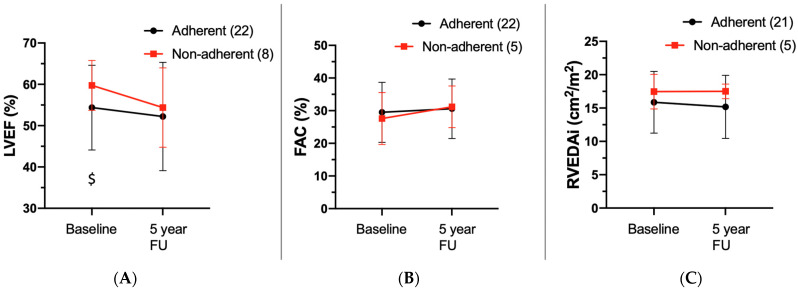
Echocardiographic findings. (**A**) Comparison of left ventricular ejection fraction (LVEF%) at baseline and follow-up. (**B**) Comparison of fractional area change (FAC%) at baseline and follow-up. (**C**) Comparison of right ventricular end-diastolic area indexed (RVEDAi cm^2^/m^2^) at baseline and follow-up. Red line indicates cohort, who continued physical activity (non-adherent); black line indicates cohort, who stopped physical activity (adherent $ indicates *p* < 0.05 for independent Student’s *t*-test between the two cohorts.

**Table 1 jcm-11-01150-t001:** Patients Characteristics.

Patient Characteristics (*n* = 49 Total)	Adherent *N* = 37	Non-Adherent *N* = 12	*p* Value
Male	24 (65%)	9 (75%)	0.515
Age at baseline (years)	45.3 ± 14.3	47.5 ± 18.0	0.723
Definite ARVC	32 (86.5%)	10 (83%)	0.786
Borderline ARVC	5 (13.5%)	2 (17%)	0.591
Genetic variant associated with ARVC	10 (27%)	2 (17%)	0.301
Systolic blood pressure (mmHg)	119 ± 11	124 ± 15	0.243
Diastolic blood pressure (mmHg)	76 ± 10	79 ± 9	0.237
Heart rate (bpm)	60 (18)	62 ± 7	0.427
Diabetes	2 (5%)	0 (0%)	0.411
Arterial hypertension	6 (16%)	2 (17%)	0.971
Previous sustained ventricular arrhythmia	15 (41%)	5 (14%)	0.332
Previous sudden cardiac arrest	5 (14%)	1 (8%)	0.634
ICD implanted	25 (68%)	9 (75%)	0.627
Sustained ventricular tachycardia during FU	16 (43%)	6 (50%)	0.683

**Table 2 jcm-11-01150-t002:** Antiarrhythmic medication.

	Baseline		Follow-Up	
	Non-adherent *N* = 12	Adherent *N* = 37	Non-adherent *N* = 12	Adherent *N* = 37
Amiodarone [*N*, %]	1 (8%)	7 (19%)	2 (17%)	7 (19%)
Betablocker [*N*, %]	3 (25%)	12 (32%)	3 (25%)	20 (54%)
Betablocker % of maximum dose [median, [IQR]]	47.5 [18.8]	25 [25.0]	12.5 [6.3]	25 [18.8]
Others [*N*, %] (sotalol, mexiletine, flecainide)	0 (0%)	9 (24%)	0 (0%)	8 (22%)

## Data Availability

Data supporting reported results will be provided by the authors upon reasonable request.

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
