# Peer review of "Changes in Exercise Capacity and Ventricular Function in Arrhythmogenic Right Ventricular Cardiomyopathy: The Impact of Sports Restriction during Follow-Up"

_jcm, 2022, doi:10.3390/jcm11051150_

Round 1
Reviewer 1 Report
Costa et al. present a relevant study to understand, if ongoing sport in ARVC would impact the outcome. I would congratulate the authors for this important study.
Minor editions are required
The study is designed well. The conclusion is clear. As I understand a part of patients received ICDs. Did you document an increase of VTs or nsVTs?
Do you have any data abot cardiac MRI at the time point of diagnosis and over follow-up (ongoing sport)
Please discuss the cellular mechanism of arrhythmias in ARVC through sport (e.g. PMID: 29566126, PMID: 23354045)
Author Response
"Please see attachment."

Reviewer 2 Report
Strength: This study focused on the role of modifiable factors (physical performance) and patients' compliance in ARVC progression, and it results are in accordance with existing data and recommendations.
Minor suggestion: Spectrum of genetic findings in ARVC patients (genes, variants, class of pathogenicity), and genotype-phenotype correlations (if any) would increase the scientific value of the study.
Author Response
"Please see attachment."

Reviewer 3 Report
Costa et al. presented an interesting set of data on the role of physical exercise in promoting disease progression in patients with arrhythmogenic right ventricular cardiomyopathy (ARVC). They aimed to investigate the exercise performance and ventricular function of ARVC patients during follow-up, while taking into account their adherence to exercise restriction recommendations. Their study had a restrospective design, and included included 49 patients who had an exercise test at baseline and after mean 4.2 years. Of the 49 ARVC patients, 27 (55%) were athletes. Of the athletes, 12 (44%) continued intensive sports activity (non-adherent), while 15 (56%) stopped intensive physical activity upon recommendation (adherent). Authors concluded that if intensive sports activities are not discontinued, exercise capacity and left ventricular 24 function of athletes with ARVC deteriorates during follow-up.
This is a very useful clinically information that helps to guide patients with this rare disease and very well written manuscript.
Although the study had a retrospective design, it would be nice to learn about cardiac magnetic resonance imaging, at least in some patients, whether there was more fibrosis that could account for deterioration of left ventricular function. At least in the Discussion section, the role of CMR or biomarkers that could shed the light on the structural remodelling of the heart in patients with ARVC should be added.
Few spelling mistakes should be corrected,
FAC (fac) is written both as FAC and fac...
fortysix
fortytwo
Author Response
"Please see attachment."
